# Risk Stratification with Sarculator and MSKCC in Patients with Primary and Secondary Angiosarcoma

**DOI:** 10.3390/life14050569

**Published:** 2024-04-28

**Authors:** Yonca Steubing, Nilofar Ahmadi, Flemming Puscz, Alexander Wolff, Jannik Hinzmann, Felix Reinkemeier, Sonja Verena Schmidt, Alexander Sogorski, Maxi Von Glinski, Mustafa Becerikli, Maria Füth, Jessica Zuchowski, Hannah Brüggenhorst, Tom Huyghebaert, Ingo Stricker, Marcus Lehnhardt, Christoph Wallner

**Affiliations:** 1Department of Plastic Surgery, Hand Surgery, Burn Center, and Sarcoma Center, BG University Hospital Bergmannsheil Bochum, 44789 Bochum, Germany; 2Institute of Pathology, Ruhr-University Bochum, 44801 Bochum, Germany

**Keywords:** angiosarcoma, Sarculator, soft tissue sarcoma, radiotherapy

## Abstract

Background: Sarculator and Memorial Sloan Kettering Cancer Center (MSKCC) nomograms are freely available risk prediction scores for surgically treated patients with primary sarcomas. Due to the rarity of angiosarcomas, these scores have only been tested on small cohorts of angiosarcoma patients. In neither the original patient cohort upon which the Sarculator is based nor in subsequent studies was a distinction made between primary and secondary angiosarcomas, as the app is intended to be applied to primary sarcomas. Therefore, the objective of our investigation was to assess whether the Sarculator reveals a difference in prognosis and whether such differentiation aligns with actual clinical data. Patients and Methods: Thirty-one patients with primary or secondary soft tissue angiosarcoma, treated at our Sarcoma Center from 2001 to 2023, were included in the study. Actual survival rates were compared with nomogram-derived data for predicted 5-year survival (Sarculator), as well as 4-, 8- and 12-year sarcoma-specific death probabilities (MSKCC). Harrell’s c-index was utilized to assess predictive validity. Results: In total, 31 patients were analyzed. The actual overall 5-year survival was 22.57% with a predicted 5-year survival rate of 25.97%, and the concordance index was 0.726 for the entire cohort. The concordance index results from MSKCC for angiosarcoma patients were below 0.7 indicating limited predictive accuracy in this cohort, particularly when compared to Sarculator. Summary: Nomogram-based predictive models are valuable tools in clinical practice for rapidly assessing prognosis. They can streamline the decision-making process for adjuvant treatments and improve patient counselling especially in the treatment of rare and complicated tumor entities such as angiosarcomas.

## 1. Introduction

The risk stratification for soft tissue sarcomas is challenging due to their rarity and heterogeneity. The free available Sarculator app was designed with the aim of simplifying and improving prognosis generation and aiding physicians in risk stratification for patients with soft tissue sarcomas [1]. It was developed by Callegaro et al. from Fondazione IRCCS Istituto Nazionale dei Tumori in Milan and Vergani from Digital Forest in Milan in 2016. The app is based on prognostic nomograms derived from patients with surgically removed soft tissue sarcomas of the extremities and trunk [2]. These data were validated using patient data from France, England, and Canada [3]. For prediction of the 5- and 10-year overall survival and metastasis-free survival rate of patients, the app requires the patient age, tumor diameter, as well as FNCLCC-grading (Fédération Nationale des Centres de Lutte Contre le Cancer) and histology. Additionally, for patients with sarcoma recurrence, Sarculator can provide the 5-year survival rates at various points in time within the first three years of the follow-up. Clinical studies demonstrated the predictive capability of survival in sarcoma patients through Sarculator [3,4]. Another validated prognostic model is the Memorial Sloan Kettering Cancer Center (MSKCC) nomogram developed by Kattan, Leung, and Brennan in 2002 [5]. Various prognostic nomograms are accessible on the clinic’s website, estimating the sarcoma-specific death (SSD) at 4, 8, or 12 years after surgical treatment or local recurrence [6]. The variables required for calculation include patient’s age, tumor location relative to the muscle fascia, grading, histology, tumor localization, and size. To assess the prognostic capabilities of the described nomograms on a patient cohort from our clinic, we compared the actual survival of angiosarcoma patients with the predicted survival by Sarculator and MSKCC. As angiosarcomas represent an extremely rare entity even among soft tissue sarcomas, limited research has been conducted in this area. At MSKCC, angiosarcomas cannot be specifically selected as a distinct histology, while Sarculator offers the “vascular” option, encompassing angiosarcomas and hemangioendotheliomas. To date, there has been no differentiation in the prognosis of primary and secondary angiosarcomas. The aim of our study was to evaluate the predictive quality of the models and investigate whether a distinction in prognosis exists between primary and secondary angiosarcomas even if it was originally designed for primary sarcoma patients.

## 2. Angiosarcoma

Angiosarcomas are a rare and aggressive form of soft tissue sarcomas with a poor prognosis. They originate from endothelial cells in blood or lymphatic vessels and occur as a primary form, especially as cutaneous in the head area, or secondary after radiation therapy, particularly in breast cancer patients or in the context of lymphedema [7] (Figure 1). Soft tissue sarcomas represent a rare tumor entity, with angiosarcomas constituting approximately 2% of the soft tissue sarcoma spectrum [8,9]. The etiology of primary angiosarcomas has not been elucidated, risk factors for secondary angiosarcomas include radiotherapy, particularly following adjuvant treatment for breast carcinoma. Another predisposing factor is chronic lymphedema, which is also often a consequence of surgical breast cancer therapy (Stewart–Treves syndrome) [10]. Post-radiation angiosarcomas of the breast are the most common radiation-associated tumors, manifesting with a latency period of 5–10 years after therapy. However, post-radiation angiosarcomas may also manifest at other anatomical sites subsequent to radiation exposure [7]. Clinically, they often present as a type of skin rash or hematoma, leading to occasional initial misdiagnosis [11]. The multifocal nature of growth necessitates a comprehensive surgical excision, representing the primary therapeutic focal point [12]. Current recommendations regarding adjuvant radiotherapy are divided. Various studies have reported survival advantages for combined surgery and subsequent radiotherapy compared to sole surgery or radiotherapy alone [13,14]. Recurrence of angiosarcomas is common, leading to a generally poor prognosis [15].

## 3. Methods

### 3.1. Patient Selection

A total of 31 patients with surgically treated primary (PAS) or secondary soft tissue angiosarcoma (SAS) at our clinic between 2001 to 2023 were identified from our database and included in the study. According to the criteria of Sarculator, patients with a tumor size >35 cm, incomplete surgical intervention (R2-resection) and those experiencing recurrence after 36 months were excluded (n = 9). Patients with visceral angiosarcomas were also excluded. For all patients, case presentations were conducted in our interdisciplinary tumor board involving oncologists, radiation therapists, pathologists, internists, and surgeons from various departments and all patients received treatment at our sarcoma center.

According to the German S3 guideline for soft tissue sarcomas, postoperative follow-up examinations were conducted with contrast-enhanced MRI of the tumor region and imaging of the lungs (two-plane chest X-ray or low-dose chest CT) every 3 months during the first 2 years postoperatively. Subsequently, follow-ups were scheduled every 6 months for the next 3 years, followed by annual assessments after 5 years [16].

### 3.2. Statistical Analysis

After identifying suitable study participants, the actual overall survival (aOS) was calculated over the period from the date of surgery (=initial diagnosis) to the date of death and censored at the last follow-up or after 5 years if the patients survived beyond this timeframe. The tumor-specific information and patient data were entered into the app and website, thereby determining the predicted overall survival (pOS) for 5 years and sarcoma-specific death probability after 4, 8 and 12 years. To evaluate the predictive performance for angiosarcoma patients of the two nomograms, Harrell’s c-index (concordance index) was determined using Python (Version 3.12.2, Python Software Foundation, Beaverton, OR, USA) along with the libraries NumPy (NumPy Developers, version 1.26.4) for survival modeling. Descriptive statistics were performed using SPSS (IBM GmbH, version 29.0, Ehningen, Germany) for data analysis. Harrell’s c-index is a measure of accuracy for predictive models that generate risk scores. It is used to assess the efficacy of risk models in survival analysis [17]. Values around 0.5 indicate predictions akin to random chance, values approaching 0.7 signify a well-performing model, and 1.0 represents a perfect predictive model [18]. For calibration, the data was divided into equally sized groups according to quartiles and Harrell’s c-index within each group was determined. Initially, this was done for all angiosarcoma patients, and subsequently, separately for primary and secondary angiosarcomas.

## 4. Results

A total of 31 angiosarcoma patients were included in the study, with 10 having primary and 21 secondary angiosarcoma. Among them, 19 patients had a history of breast cancer and radiation therapy. The mean follow-up period was 56.44 months (range 3.25–260.13 months). Mean patient age was 65.97 years, with those having primary angiosarcoma being on average 8.81 years younger than those with secondary angiosarcoma (60 years vs. 68.81 years). The mean tumor size was 9.35 cm. A total of 67.7% of angiosarcomas were completely surgically resected (R0) and in 32.3% of cases, residual tumor cells were detected at the resection margin (R1 resection). In some patients, achieving a R0 resection was not possible due to anatomical constraints and the extent of the tumor. Adjuvant radiotherapy was administered to 35.5% of patients, while 16.1% received adjuvant chemotherapy. A total of 58.1% of patients experienced a local recurrence, with a mean time to recurrence of 372.3 days (corresponding to disease-free survival). The percentage of recurrences and the time to recurrence were similar for primary and secondary angiosarcomas (PAS vs. SAS: 60% and 376.82 days vs. 57.1% and 370.1 days). In the observation period, 20 out of 31 patients deceased and 11 patients were alive at the time of the last follow-up (Table 1).

The mean actual overall survival for all angiosarcoma patients was 56.42 months (4.7 years), and the 5-year actual overall survival was 22.57% with a predicted overall survival of 25.97%. The c-index for all angiosarcoma patients was 0.726. The 4-year sarcoma specific death (SSD) probability was 37.94% with a c-index of 0.597 compared to the actual survival. The data for the 8- and 12-year SSD-probabilities were not presented, as the values were similar to the 4-year values.

For PAS patients, the actual mean overall survival was 85 months (7.05 years), with a 5-year survival rate of 30%, while the predicted overall 5-year survival from Sarculator was 30.7%, with a c-index of 0.744. The 4-year SSD was 37.1% with a c-index of 0.678.

For SAS patients, the mean actual overall survival was 42.98 months (3.58 years) with a 5-year aOS of 19.05% and predicted survival of 23.7%, with a c-index of 0.719. The 4-year SSD was 38.32% with a c-index of 0.565.

The actual and predicted 5-year overall survival exhibited strong concordance as indicated by the c-index. This alignment was particularly notable in the lower percentiles of the study population, as depicted in Figure 2. The Kaplan-Meier survival analyses demonstrate the influence of tumor depth, indicating a significantly worse prognosis for subfascial angiosarcomas compared to superficial manifestations (see Figure 3).

## 5. Discussion

The c-index values for the Italian cohort, upon which the Sarculator nomogram is based, were reported as 0.767 (range 0.698 to 0.775) [2]. Our results confirmed the predictive accuracy of the Sarculator, as our concordance indices exceeded 0.7, indicative of a valid predictive model [18]. In a study by Voss et al. involving sarcoma patients in the USA, c-indices for angiosarcoma patients were determined to be 0.696, with an actual overall survival of 47.99% and a predicted overall survival of 40.80% [3]. In our study, the predicted overall survival (POS) of 25.97% was markedly lower, which may account for the more accurate c-index.

At MSKCC, the classification of angiosarcomas is subsumed under the general category of ‘other’ [6]. In our patient cohort, the model exhibited less robust discriminative capabilities compared to Sarculator. For primary angiosarcomas, a concordance index of 0.678 was achieved, with the remaining indices being lower. The fact that the histology categorized as ‘other’ does not offer suitable prognostic insights for angiosarcomas further underscores the poor prognosis associated with angiosarcomas when compared to other sarcoma entities. It is noteworthy that the MSKCC nomogram, developed in 2002, predates subsequent revisions in the WHO classification of soft tissue and bone tumors. Recognizing these alterations, Callegaro et al. subsequently developed a nomogram aligned with the contemporary classification [2].

An inherent limitation of this study is the relatively small sample size, a characteristic common to investigations on angiosarcomas given their infrequency. Notably, even within the Italian patient cohort serving as the foundation for the Sarculator, only 35 cases of vascular sarcoma were identified, encompassing both angiosarcoma and hemangioendothelioma [2].

Even in our small patient cohort, the Sarculator demonstrates a better predictive ability for primary (0.744) compared to secondary angiosarcomas (0.719) and the overall cohort (0.726). Prognostically, primary angiosarcomas show significantly better outcomes in larger cohorts than post-radiogenic secondary angiosarcomas. Particularly for angiosarcomas, it would be advisable to include a variable ‘radiation’ in the application [19].

The prognostic outcomes generated by the Sarculator are restricted to patients who have undergone a complete macroscopic surgical resection (R0 and R1). Different recommendations exist regarding the surgical therapy for angiosarcomas. In the Sarculator, the safety margin does not play a decisive role, yet valid prognostic data are generated, which suggests that safety margins of the resection borders may play a secondary role in influencing outcomes. Other studies, however, demonstrate a higher rate and faster occurrence of local recurrences with inadequate safety margins, such as R1 resection [20]. Furthermore, patients presenting with disease recurrence exhibit a diminished overall survival prognosis in comparison to individuals with local control or primary sarcoma [21,22]. The majority of study results indicate that complete surgical resection (R0) is associated with a better prognosis, with the extent of the surgical margin appearing to play a minor role [23,24]. In a comparative study by Li et al., an improved prognosis was observed for patients undergoing a more radical operation involving excision of the entire irradiated field compared to patients with limited resection of post-radiogenic angiosarcoma [22]. Due to the lower rate of local recurrences, an increased disease-specific survival was demonstrated. Another challenging characteristic of post-radiogenic angiosarcomas is their multifocality, which complicates the determination of a resection margin for pathologists and even more advocates for a more radical resection of the irradiated area. In total, the therapy of angiosarcomas is intricate, with a lack of larger studies investigating the extent of resection in surgical treatment, as well as recommendations for adjuvant radiation and chemotherapy. Currently, achieving better local control of post-radiogenic angiosarcomas through a radical surgical approach appears to be the most effective treatment.

The genetic characteristics of tumor subtypes significantly influence patient prognosis by determining the aggressiveness of the tumors. With advancements in molecular genetic techniques, such as Next Generation Sequencing, various approaches including three-dimensional tumor models are being explored for the testing of pharmacological therapies. These models aim to facilitate the development of targeted immunotherapies in the future. The heterogeneity of tumor subtypes presents a significant challenge in this context [25].

In clinical decision-making, nomogram-based predictive tools prove to be valuable instruments for assessing the mortality and recurrence probabilities in sarcoma patients. Informed by the predicted data, the urgency and extent of postoperative adjuvant therapies could be influenced. However, their effectiveness relies on regular updates, considering the ongoing evolution and improvement of patient therapies [26]. It is noteworthy that the Sarculator and MSKCC pertain specifically to postoperative scenarios and do not serve as predictive tools for preoperative consultations [5]. Additionally, it is essential to acknowledge that these nomograms exclusively represent disease-specific survival probabilities and do not encompass other potential causes of mortality.

## Figures and Tables

**Figure 1 life-14-00569-f001:**
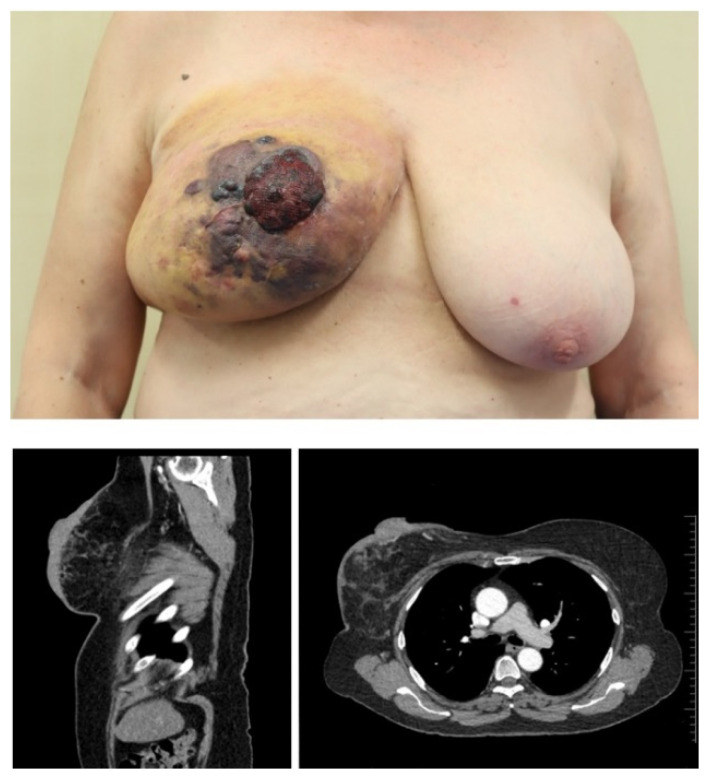
Clinical images (**top**) of a patient with post-radiogenic angiosarcoma, and corresponding CT thoracic images (**bottom**) in two planes.

**Figure 2 life-14-00569-f002:**
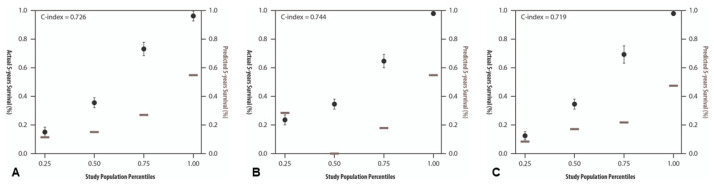
Calibration plots of actual 5-year overall survival (left y-axis, black boxplots) against predicted 5-year overall survival (right y-axis, brown vertical bars) of the Sarculator in percentage. The x-axis shows the percentiles of the study population. (**A**) Actual and predicted 5-year overall survival for the complete patient cohort, (**B**) for primary and (**C**) for secondary angiosarcoma patients.

**Figure 3 life-14-00569-f003:**
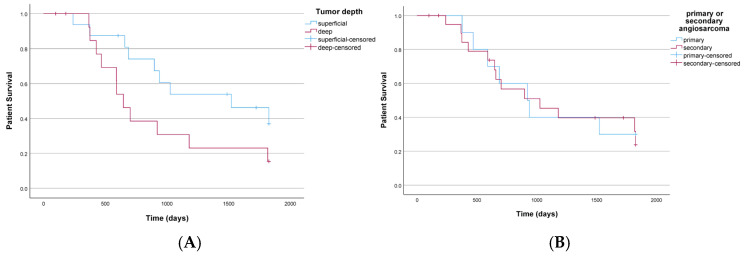
Kaplan–Meier survival analysis: (**A**) illustrates patient survival (x-axis) relative to time in days (y-axis) stratified by tumor depth. The blue curve represents patients with tumors located above the muscle fascia, while the red curve denotes subfascial location. (**B**) Depicts Kaplan–Meier curves for patients with primary angiosarcoma (blue) and secondary angiosarcoma (red). The vertical lines indicate censored data.

**Table 1 life-14-00569-t001:** Demographic Characteristics (patient, tumor and treatment).

Patient and Tumor Characteristics	n = 31
Primary Angiosarcoma	10 (32.4%)
Secondary Angiosarcoma	21 (67.6%)
Median age (years)	65.97 (IQR 50–74, range 38–86)
Sex (men/women)	5/26
Breast Carcinoma in medical history	19 (61.3%)
Median tumor size (cm)	9.35 (IQR 4.13–18.88, range 0.5–30)
Subfascial	15 (48.4%)
Histology	1 (hemangioendothelioma), 4 (primary cutaneous angiosarcoma), 5 (primary soft tissue angiosarcoma), 21 (postradiogenic angiosarcomas)
Grading	2 (G1), 3 (G2), 26 (G3)
Median follow up (months)	56.44 (range 3.25–260.13)
Median disease-free Survival (months)	12 (IQR 12.3–78.62, range 2.12–31.42)
Death during study period	20 (64.4%)
**Treatment characteristics**	
R0	21 (67.6%)
R1	10 (32.4%)
Adjuvant Chemotherapy	5 (16%)
Adjuvant Radiation Therapy	11 (35.5%)
Local Recurrence	18 (58%)

Values are given as mean with corresponding interquartile range (IQR) or n (% of group).

## Data Availability

The data presented in this study are available on request from the corresponding author. The data are not publicly available due to the sensitive nature of patient data, which were only utilized in pseudonymized form for the purpose of this retrospective study.

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
