# Peer review of "Risk Stratification with Sarculator and MSKCC in Patients with Primary and Secondary Angiosarcoma"

_life, 2024, doi:10.3390/life14050569_

Round 1

Reviewer 1 Report

Comments and Suggestions for Authors

 Manuscript ID: life-2933749

Title: Risk stratification with Sarculator and MSKCC in patients with primary and secondary angiosarcoma

In this manuscript, the authors test models to predict survival in patients with angiosarcoma.

The data analysis section requires some clarification. The main weakness is that is that the conclusions are based on a relatively small number, and then the use of statistical analysis of data reduces the degree of confidence in the results.

The study indicates the tests performed but it simply compares two models.

The authors should add, in the discussion, as the genomic and epigenomic characteristics of tumor subtypes influence the risk, prognosis and treatment (please see: https://doi.org/10.3390/cells12172204)

Comments on the Quality of English Language

Manuscript ID: life-2933749

Title: Risk stratification with Sarculator and MSKCC in patients with primary and secondary angiosarcoma

In this manuscript the authors test models to predict survival in patients with angiosarcoma.

The data analysis section requires some clarification. The main weakness is that is that the conclusions are based on a relatively small number, and then the use of statistical analysis of data reduces the degree of confidence in the results.

The study indicates the tests performed but it simply compares two models.

The authors should add, in discussion, as the genomic and epigenomic characteristics of tumor subtypes influence the risk, prognosis and treatment (please see: https://doi.org/10.3390/cells12172204)

Author Response

Dear Reviewer,

Thank you very much for your insightful comments and suggestions. We appreciate the time and effort you have taken to review our manuscript and believe that your feedback has greatly contributed to improving the quality of our paper.

1) Data analysis: 

The data analysis section indeed warrants clarification, with a notable weakness stemming from the limited sample size, exacerbated by the rarity of angiosarcomas even among rare soft tissue sarcomas. Consequently, achieving a large case count, even within a comprehensive sarcoma center such as ours, is challenging. Multicenter studies present a potential solution; however, they encounter hurdles such as coordinating between sites, challenges in data acquisition due to the rarity of the condition, ensuring consistency in procedures and protocols across diverse centers, managing elevated costs associated with multicenter collaborations, and addressing potential heterogeneity in patient populations across sites.

In light of these constraints, the study opted for a comparative analysis of two models after both models were tested with our patient data. Despite the small sample size, statistical analyses were employed to extract meaningful insights. Notably, the Harrell's C index, a commonly utilized metric in similar studies (Wong et al. 2020; Voss et al. 2022), underwent internal validation and was deemed appropriate for this analysis. While acknowledging the limitations posed by the sample size and statistical analyses, the study endeavors to provide valuable insights within the constraints of the available data.

2) Genomic and epigenomic characteristics of tumor subtypes: 

I have added a section on this topic in the discussion:

“The genetic characteristics of tumor subtypes significantly influence patient prognosis by determining the aggressiveness of the tumors. With advancements in molecular genetic techniques, such as Next Generation Sequencing, various approaches including three-dimensional tumor models are being explored for the testing of pharmacological therapies. These models aim to facilitate the development of targeted immunotherapies in the future. The heterogeneity of tumor subtypes presents a significant challenge in this context.”

I have attached the revised manuscript below.

Best regards.

Bibliography:

Voss, Rachel K.; Callegaro, Dario; Chiang, Yi-Ju; Fiore, Marco; Miceli, Rosalba; Keung, Emily Z. et al. (2022): Sarculator is a Good Model to Predict Survival in Resected Extremity and Trunk Sarcomas in US Patients. In: Annals of surgical oncology. DOI: 10.1245/s10434-022-11442-2.

Wong, Ru Xin; Koh, Yen Sin; Ong, Faith; Farid, M.; Tay, Timothy Kwang Yong; Teo, Melissa (2020): Applicability of the Sarculator and MSKCC nomograms to retroperitoneal sarcoma prognostication in an Asian tertiary center. In: Asian journal of surgery 43 (11), S. 1078–1085. DOI: 10.1016/j.asjsur.2020.01.005.

Reviewer 2 Report

Comments and Suggestions for Authors

Manuscript entitled "Risk stratification with Sarculator and MSKCC in patients with primary and secondary angiosarcoma"

This work is generally well organized while there are some issues to be modified:

1. The cases should be described in more detail regarding the histopathological features.

2. Representative images of selected cases, including radiographic and pathologic, should be posted.

3. A K-M curve should be provided for cases with different categories, ex. primary vs secondary; trunk vs periphery; deep vs superficial.

Comments on the Quality of English Language

Acceptable

Author Response

Dear Reviewer,

Thank you very much for your insightful comments and suggestions. We appreciate the time and effort you have taken to review our manuscript and believe that your feedback has greatly contributed to improving the quality of our paper.

1) The cases should be described in more detail regarding the histopathological features.

  • Histopathological characteristics regarding tumor entity as well as grading were added to the demographics table (see attachment).

2) Representative images of selected cases, including radiographic and pathologic, should be posted.

  • Representative clinical and CT images of a patient were included (see attachment); however, images of pathological stains were not obtained as they are not standard practice in our institution.

3) A K-M curve should be provided for cases with different categories, ex. primary vs secondary; trunk vs periphery; deep vs superficial.

  • As requested, I have provided two Kaplan-Meier curves for primary vs. secondary and tumor depth superficial vs. deep. However, no informative graphs were generated for trunk vs. extremities or for, for example, a history of breast carcinoma.

I have attached the revised manuscript along with the images, CT scans, and Kaplan-Meier curves below, as only one file can be attached.

Best regards.

Reviewer 3 Report

Comments and Suggestions for Authors

This research paper compares MSKCC and Sarculator in terms of prognostic prediction of angiosarcoma patients. This is an interesting paper, but the following points require additional correction.

#1. It is known that the prognosis of angiosarcoma differs significantly between cutaneous and visceral cases. In the present study, the cases are classified as primary and secondary, but what are the results if a distinction is made between cutaneous and visceral cases?

#2. As the authors themselves state, the number of cases is too small, which reduces the value of this paper. Would it be difficult to increase the number of cases in the form of a multicentre study?

#3. Was this study approved by IRB? A statement on this point is needed.

Comments on the Quality of English Language

Minor editing is required.

Author Response

Dear Reviewer,

Thank you very much for your insightful comments and suggestions. We appreciate the time and effort you have taken to review our manuscript and believe that your feedback has greatly contributed to improving the quality of our paper.

1) It is known that the prognosis of angiosarcoma differs significantly between cutaneous and visceral cases. In the present study, the cases are classified as primary and secondary, but what are the results if a distinction is made between cutaneous and visceral cases?

  • As a department for plastic surgery, our patient cohort primarily comprised cases of cutaneous and soft tissue angiosarcomas. Patients with visceral angiosarcomas were excluded from this study. A clarifying statement regarding this exclusion was deemed essential and has been added to the "Patient Selection" subsection under the Methods.

2) As the authors themselves state, the number of cases is too small, which reduces the value of this paper. Would it be difficult to increase the number of cases in the form of a multicentre study?

  • Enhancing the case count via a multicenter study poses challenges due to the notably low nationwide incidence of angiosarcomas within Germany, such as coordinating between sites, challenges in data acquisition, ensuring consistency in procedures and protocols, managing higher costs, and addressing potential heterogeneity in patient populations across sites.

3) Was this study approved by IRB? A statement on this point is needed.

  • According to our responsible ethics committee of the Medical Association Westphalia-Lippe, no ethics approval is necessary for this study. Attached is a section from the website for your reference.

“Exception to the necessity of consent

According to § 15 (1) of the Professional Code of the Medical Association Westphalia-Lippe in conjunction with item 1 of the Declaration of Helsinki by the World Medical Association (Rev. 2013), consultation with the ethics committee is also necessary for retrospective analysis of clinical routine data if it does not exclusively concern anonymized (i.e., personal/pseudonymized) data. Additionally, most publishing bodies in this case also require an appropriate ethics approval.”

https://www.aekwl.de/fuer-aerzte/ethik-kommission/antragsunterlagen-hilfestellung/retrospektive-studien (website in german)

Since the data originate retrospectively from routine examinations and are utilized in anonymized form, as per the guidelines of our ethics committee, there is no requirement for an ethics approval. For precise specification, I have added the following paragraph in the back matter: 

"Institutional Review Board Statement: Ethical review and approval were waived for this study due to the retrospective nature of the data originating from routine examinations, which are utilized in anonymized form, in accordance with the guidelines provided by our responsible ethics committee of the Medical Association Westphalia-Lippe."

I have attached the revised manuscript below.

Best regards.

Round 2

Reviewer 2 Report

Comments and Suggestions for Authors

The revision is acceptable in the present form.

Reviewer 3 Report

Comments and Suggestions for Authors

The authors have appropriately responded to my request.